# A Representation Theory for Ranking Functions

**Harsh Pareek, Pradeep Ravikumar**
Department of Computer Science
University of Texas at Austin
{harshp,pradeepr}@cs.utexas.edu

## Abstract

This paper presents a representation theory for permutation-valued functions, which in their general form can also be called *listwise ranking functions*. Pointwise ranking functions assign a score to each object independently, without taking into account the other objects under consideration; whereas listwise loss functions evaluate the set of scores assigned to all objects as a whole. In many supervised learning to rank tasks, it might be of interest to use listwise ranking functions instead; in particular, the Bayes Optimal ranking functions might themselves be listwise, especially if the loss function is listwise. A key caveat to using listwise ranking functions has been the lack of an appropriate representation theory for such functions. We show that a natural symmetricity assumption that we call *exchangeability* allows us to explicitly characterize the set of such exchangeable listwise ranking functions. Our analysis draws from the theories of tensor analysis, functional analysis and De Finetti theorems. We also present experiments using a novel reranking method motivated by our representation theory.

## 1 Introduction

A permutation-valued function, also called a ranking function, outputs a ranking over a set of objects given features corresponding to the objects, and learning such ranking functions given data is becoming an increasingly key machine learning task. For instance, tracking a set of objects given a particular order of uncertain sensory inputs involves predicting the permutation of objects corresponding to the inputs at each time step. Collaborative filtering and recommender systems can be modeled as ranking movies (or other consumer objects). Extractive document summarization involves ranking sentences in order of their importance, while also taking diversity into account. Learning rankings over documents, in particular, has received considerable attention in the Information Retrieval community, under the subfield of "learning to rank". The problems above involve diverse kinds of supervision and diverse evaluation metrics, but with the common feature that the object of interest is a ranking function, that when given an input set of objects, outputs a permutation over the set of objects. In this paper, we will consider the standard generalization of ranking functions which output a real-valued score vector, which can be sorted to yield the desired permutation.

The tasks above then entail learning a ranking function given data, and given some evaluation metric which captures the compatibility between two permutations. These evaluation metrics are domain-specific, and even in specific domains such as information retrieval, could be varied based on actual user preferences. Popular IR evaluation metrics for instance include Mean Average Precision (MAP) [1], Expected Reciprocal Rank (ERR) [7] and Normalized Discounted Cumulative Gain (NDCG) [17]. A common characteristic of these evaluation loss functionals are that these are typically *listwise*: so that the loss evaluates the entire set of scores assigned to all the objects in a manner that is not separable in the individual scores. Indeed, some tasks by their very nature require listwise evaluation metrics. A key example is that of *ranking with diversity*[5], where the user prefers results that are not only relevant individually, but also diverse mutually; searching for web-pages with the query "Jaguar" should not just return individually relevant results, but also results that cover

the car, the animal and the sports team, among others. Chapelle et al [8] also mention ranking for diversity as an important future direction in learning to rank. Other fundamentally listwise ranking problems include pseudo-relevance feedback, topic distillation, subtopic retrieval and ranking over graphs (e.g.. social networks) [22].

While these evaluation/loss functionals (and typically their corresponding surrogate loss functionals as well) are listwise, most parameterizations of the *ranking functions* used within these (surrogate) loss functionals are typically *pointwise*, i.e. they rank each object (e.g. document) independently of the other objects. Why should we require listwise ranking functions for listwise ranking tasks? Pointwise ranking functions have the advantage of computational efficiency: since these evaluate each object independently, they can be parameterized very compactly. Moreover, for certain ranking tasks, such as vanilla rank prediction with $0/1$ loss or multilabel ranking with certain losses[11], it can be shown that the Bayes-consistent ranking function is pointwise, so that one would lose statistical efficiency by not restricting to the sub-class of pointwise ranking functions. However, as noted above, many modern ranking tasks have an inherently listwise flavor, and correspondingly their Bayes-consistent ranking functions are listwise as well. For instance, [24] show that the Bayes-consistent ranking function of the popular NDCG evaluation metric is inherently listwise.

There is however a caveat to using listwise ranking functions: a lack of representation theory, and corresponding guidance to parameterizing such listwise ranking functions. Indeed, the most commonly used ranking functions are linear ranking functions and decision trees, both of which are *pointwise*. With decision trees, gradient boosting is often used as a technique to increase the complexity of the function class. The Yahoo! Learning to Rank challenge [6] was dominated by such methods, which comprise the state-of-the-art in learning to rank for information retrieval today. It should be noted that gradient boosted decision trees, even if trained with listwise loss functions (e.g.. via LambdaMART[3]), are still a sum of pointwise ranking functions and therefore pointwise ranking functions themselves, and hence subject to the theoretical limitations outlined in this paper.

In a key contribution of this paper, we impose a very natural assumption on general listwise ranking functions, which we term *exchangeability*, which formalizes the notion that the ranking function depends only on the object features, and not the order in which the documents are presented. Specifically, as detailed further in Section 3, we define *exchangeable* ranking functions as those listwise functions where if their set of input objects is permuted, their output permutation/score vector is permuted in the same way. This simple assumption allows us to provide an *explicit characterization* of the set of listwise ranking functions in the following form:

$$(\mathbf{f}(\mathbf{x}))_i = h(\mathbf{x}_i, \{\mathbf{x}_{\setminus i}\}) = \sum_t \Pi_{j \neq i} g_t(\mathbf{x}_i, \mathbf{x}_j) \tag{1}$$

This representation theorem is the principal contribution of this work. We hope that this result will provide a general recipe for designing learning to rank algorithms for diverse domains. For each domain, practitioners would need to utilize domain knowledge to define a suitable class of pairwise functions $g$ parameterized by $w$, and use this ranking function in conjunction with a suitable listwise loss. Individual terms in (1) can be fit via standard optimization methods such as gradient descent, while multiple terms can be fit via gradient boosting.

In recent work, two papers have proposed specific listwise ranking functions. Qin et al. [22] suggest the use of conditional random fields (CRFs) to predict the relevance scores of the individual documents via the the most probable configuration of the CRF. They distinguish between "local ranking," which we called ranking with pointwise ranking functions above, and "global ranking" which corresponds to listwise ranking functions; and argue that using CRFs would allow for global ranking. Weston and Blitzer [26] propose a listwise ranking function ("Latent Structured Ranking") assuming a low rank structure for the set of items to be ranked. Both of these ranking functions are exchangeable as we detail in Appendix A. The improved performance of these specific classes of ranking functions also provides empirical support for the need for a representation theory of general listwise ranking functions.

We first consider the case where features are discrete and derive our representation theorem using the theory of symmetric tensor decomposition. For the more general continuous case, we first present the the case with three objects using functional analytic spectral theory. We then present the extension to the general continuous case by drawing upon De Finetti's theorem. Our analysis highlights the correspondences between these theories, and brings out an important open problem in the functional analysis literature.

## 2 Problem Setup

We consider the general ranking setting, where the $m$ objects to be ranked (possibly contingent on a query), are represented by the feature vectors $\mathbf{x} = (\mathbf{x}_1, \mathbf{x}_2, \ldots, \mathbf{x}_m) \in \mathcal{X}^m$. Typically, $\mathcal{X} = \mathbb{R}^k$ for some $k$. The key object of interest in this paper is a ranking function:

**Definition 2.1 (Ranking function)** *Given a set of object feature vectors* $\mathbf{x}$ *(possibly contingent on a query $q$), a ranking function $f : \mathcal{X}^m \to \mathbb{R}^m$ is a function that takes as input the $m$ object feature vectors, and has as output a vector of scores for the set of objects, so that $f(\mathbf{x}) = (f_1(\mathbf{x}), \ldots, f_m(\mathbf{x}))$; for some functions $f_j : \mathcal{X}^m \to \mathbb{R}$.*

It is instructive at this juncture to distinguish between pointwise (local) and listwise (global) ranking functions. A *pointwise* ranking function $f$ would score each object $\mathbf{x}_i$ independently, ignoring the other objects, so that each component function $f_j(\mathbf{x})$ above depends only on $\mathbf{x}_j$, and can be written as a function $f_j(\mathbf{x}_j)$ with some overloading of notation. In contrast, the components $f_j(\mathbf{x})$ of the output vector of a *listwise* ranking function would depend on the feature-vectors of all the documents.

## 3 Representation theory

We investigate the class of ranking functions which satisfy a very natural property: exchanging the feature-vectors of any two documents should cause their positions in the output ranking order to be exchanged. Definition 3.1 formalizes this intuition.

**Definition 3.1 (Exchangeable Ranking Function)** *A listwise ranking function $f : \mathcal{X}^m \to \mathbb{R}^m$ is said to be exchangeable if $f(\pi(\mathbf{x})) = \pi(f(\mathbf{x}))$ for every permutation $\pi \in \mathcal{S}_k$ (where $\mathcal{S}_k$ is the set of all permutations of order $k$)*

Letting $(f_1, f_2, \ldots, f_m)$ denote the components of the ranking function $f$, we arrive at the following key characterization of exchangeable ranking functions.

**Theorem 3.2** *Every exchangeable ranking function $f : \mathcal{X}^m \to \mathbb{R}^m$ can be written as $f(\mathbf{x}) = (f_1(\mathbf{x}), f_2(\mathbf{x}), \ldots, f_m(\mathbf{x}))$ with*

$$f_i(\mathbf{x}) = h(\mathbf{x}_i, \{\mathbf{x}_{\backslash i}\}) \tag{2}$$

*where $\{\mathbf{x}_{\backslash i}\} = \{\mathbf{x}_j | 1 \leq j \leq m, j \neq i\}$, and for some $h : \mathcal{X}^m \to \mathbb{R}$ symmetric in $\{\mathbf{x}_{\backslash i}\}$ (i.e. $h(\mathbf{y}) = h(\pi(\mathbf{y})), \forall \mathbf{y} \in \mathcal{X}^{m-1}, \pi \in \mathcal{S}_k$)*

**Proof** The components of a ranking function $f : \mathcal{X}^m \to \mathbb{R}^m$, viz. $f_i(\mathbf{x})$, represent the score assigned to each document. First, exchangeability implies that exchanging the feature values of some two documents does not affect the scores of the remaining documents, i.e. $f_i(\mathbf{x})$ does not change if $i$ is not involved in the exchange, i.e. $f_i(\mathbf{x})$ is symmetric in $\{\mathbf{x}_{\backslash i}\}$ Second, exchanging the feature values of documents 1 and $i$ exchanges their scores, i.e.,

$$f_i(\mathbf{x}_1, \ldots, \mathbf{x}_i, \ldots, \mathbf{x}_n) = f_1(\mathbf{x}_i, \ldots, \mathbf{x}_1, \ldots, \mathbf{x}_n) \tag{3}$$

Thus, the scoring function for the $i$th document can be expressed in terms of that of the first document. Call that scoring function $h$. Then, combining the two properties above, we have,

$$f_i(\mathbf{x}) = h(\mathbf{x}_i, \{\mathbf{x}_{\backslash i}\}) \tag{4}$$

where $h$ is symmetric in $\{\mathbf{x}_{\backslash i}\}$. ∎

Theorem 3.2 entails the intuitive result that the *component functions $f_i$* of exchangeable ranking functions $f$ can all be expressed in terms of a single *partially symmetric* function $h$ whose first argument is the document corresponding to that component and which is symmetric in the other documents. Pointwise ranking functions then correspond to the special case where $h$ is independent of the other document-feature-vectors (so that $h(\mathbf{x}_i, \{\mathbf{x}_{\backslash i}\}) = h(\mathbf{x}_i)$ with some overloading of notation) and are thus trivially exchangeable.

As the main result of this paper, we will characterize the class of such partially symmetric functions $h$, and thus the set of exchangeable listwise ranking functions, for various classes $\mathcal{X}$ as

$$f_i(\mathbf{x}) = \sum_{t=1}^{\infty} \Pi_{j \neq i} g_t(\mathbf{x}_i, \mathbf{x}_j) \tag{5}$$

for some set of functions $\{g_t\}_{t=1}^{\infty}$, $g_t : \mathcal{X} \times \mathcal{X} \to \mathbb{R}$.

## 3.1 The Discrete Case: Tensor Decomposition

We first consider a decomposition theorem for symmetric tensors, and then through a correspondence between symmetric tensors and symmetric functions with finite domains, derive the corresponding decomposition for symmetric functions. We then simply extend the analysis to obtain the corresponding decomposition theorem for partially symmetric functions.

The term tensor may have connotations (from its use in Physics) with regards to how a quantity behaves under linear transformations, but here we use it only to mean "multi-way array".

**Definition 3.3 (Tensor)** *A real-valued order-$k$ tensor is a collection of real-valued elements $A_{i_1,i_2,\ldots,i_k} \in \mathbb{R}$ indexed by tuples $(i_1, i_2, \ldots, i_k) \in \mathcal{X}^k$.*

**Definition 3.4 (Symmetric tensor)** *An order-$k$ tensor $A = [A_{i_1,i_2\ldots,i_k}]$ is said to be symmetric iff for any permutation $\pi \in \mathcal{S}_k$,*

$$A_{i_1,i_2,\ldots,i_k} = A_{i_{\pi(1)},i_{\pi(2)},\ldots,i_{\pi(k)}}. \tag{6}$$

Comon et al. [9] show that such a symmetric tensor (sometimes called supersymmetric since it is symmetric w.r.t. all dimensions) can be *decomposed* into a sum of rank-1 symmetric tensors, where a rank-1 symmetric tensor is a $k$-way outer product of some vector $\mathbf{v}$ (we will use the standard notation $\otimes$ to denote an outer product $\mathbf{u} \otimes \mathbf{v} \otimes \cdots \otimes \mathbf{z} = [u_{j_1} v_{j_2} \ldots z_{j_k}]_{j_1,\ldots,j_k}$).

**Proposition 3.5 (Decomposition theorem for symmetric tensors [9])** *Any order-$k$ symmetric tensor $A$ can be decomposed as a sum of $k$-fold outer product tensors as follows:*

$$A = \sum_{i=1}^{\infty} \otimes^k \mathbf{v}_i \tag{7}$$

The special matrix case ($k = 2$) of this theorem should be familiar to the reader as the spectral theorem. In that case, the $\mathbf{v}_i$ are orthogonal, the smallest such representation is unique and can be recovered by tractable algorithms. In the general symmetric tensor case, the $\mathbf{v}_i$ are not necessarily orthogonal and the decomposition need not be unique; it is however finite [9]. While the spectral theory for symmetric tensors is relatively straightforward, bearing similarity to that for matrices, the theory for general non-symmetric tensors is nontrivial: we refer the interested reader to [21, 20, 10]. However, since we are interested not in general non-symmetric tensors, but partially symmetric tensors, the above theorem can be extended in a straightforward way in our case as we shall see in Theorem 3.7.

Our next step involves generalizing the earlier proposition to multivariate symmetric functions by representing them as tensors, which then yields a corresponding spectral theorem of product decompositions for such functions. In particular, note that when the feature vector of each document takes values only from a finite set $\mathcal{X}$, of size $|\mathcal{X}|$, a symmetric function $h(\mathbf{x}_1, \mathbf{x}_2, \ldots, \mathbf{x}_m)$ can be represented as an order-$m$ symmetric tensor $H$ where $H_{\mathbf{v}_1\mathbf{v}_2\ldots\mathbf{v}_m} = h(\mathbf{v}_1, \mathbf{v}_2, \ldots, \mathbf{v}_m)$ for $\mathbf{v}_i \in \mathcal{X}$. We can thus leverage Proposition 3.5 to obtain the result of the following proposition:

**Proposition 3.6 (Symmetric Product decomposition for multivariate functions (finite domain))** *Any symmetric function $f : \mathcal{X}^m \to \mathbb{R}$ for a finite set $\mathcal{X}$ can be decomposed as*

$$f(\mathbf{x}) = \sum_{t=1}^{\infty} \Pi_j g_t(\mathbf{x}_j), \tag{8}$$

*for some set of functions $\{g_t\}_{t=1}^{T}$, $g_t : \mathcal{X} \to \mathbb{R}$, $T < \infty$*

In the case of ranking three documents, each $f_i$ assigns a score to document $i$ taking the other document's features as arguments. $f_i$ then corresponds to a matrix and the functions $g_t$ correspond to the set of eigenvectors of this matrix. In the general case of ranking $m$ documents, $f_i$ is an order $m - 1$ tensor and $g_t$ are the eigenvectors for a symmetric decomposition of the tensor.

Our class of exchangeable ranking functions corresponds to *partially* symmetric functions. In the following, we extend the theory above to the partially symmetric case (proof in Appendix B).

**Theorem 3.7  (Product decomposition for partially symmetric functions)** *A partially symmetric function $h : \mathcal{X}^m \to \mathbb{R}$ symmetric in $\mathbf{x}_2, \dots, \mathbf{x}_m$ on a finite set $\mathcal{X}$ can be decomposed as*

$$h(\mathbf{x}_1, \{\mathbf{x}_{\backslash 1}\}) = \sum_{t=1}^{\infty} \Pi_{j \neq 1} g_t(\mathbf{x}_1, \mathbf{x}_j) \tag{9}$$

*for some set of functions $\{g_t\}_{t=1}^{T}, g_t : \mathcal{X} \times \mathcal{X} \to \mathbb{R}, T < \infty$.*

**Remarks:**

**I.** To the best of our knowledge, the study of *partially symmetric tensors* and their decompositions as above has not been considered in the literature. Notions such as rank and best successive approximations would be interesting areas for future research.

**II.** The tensor view of learning to rank gives rise to a host of other interesting research directions. Consider the learning to rank problem: each training example corresponds to one entry in the resulting ranking tensor. A candidate approach to learning to rank might thus be tensor-completion, perhaps using a convex nuclear tensor norm regularization [14].

## 3.2   The Continuous Case

In this section, we generalize the results of the previous section to the more realistic setting where the feature space $\mathcal{X}$ is compact. The extension to the partially symmetric case from the symmetric one is similar to that in the discrete case and is given as Theorem C.1 in Appendix C, so we discuss only decomposition theorems for symmetric functions below.

### 3.2.1   Argument via Functional Analytic Spectral Theorem

We first recall some key definitions from functional analysis [25, pp.203]. A linear operator $T$ is bounded if its norm $\|T\| = \sup_{\|x\|=1} \|Tx\|$ is finite. A bounded linear operator $T$ is self-adjoint if $T = T^*$, where $T^*$ is the adjoint operator. A linear operator $A$ from a Banach space $\mathcal{X}$ to a Banach space $\mathcal{Y}$ is compact if it takes bounded sets in $\mathcal{X}$ into relatively compact sets (i.e. whose closure is compact) in $\mathcal{Y}$.

The Hilbert-Schmidt theorem [25] provides a spectral decomposition for such compact self-adjoint operators. Let $A$ be a compact self-adjoint operator on a Hilbert space $\mathcal{H}$. Then, by the Hilbert-Schmidt theorem, there is a complete orthonormal basis, $\{\phi_n\}$, for $\mathcal{H}$ so that $A\phi_n = \lambda_n \phi_n$ and $\lambda_n \to 0$ as $n \to \infty$. $A$ can then be written as:

$$A = \sum_{n=1}^{\infty} \lambda_n \phi_n \langle \phi_n, \cdot \rangle. \tag{10}$$

We refer the reader to [25] for further details. The compactness condition can be relaxed to boundedness, but in that case a discrete spectrum $\{\lambda_n\}$ does not exist, and is replaced by a measure $\mu$, and the summation in the Hilbert-Schmidt theorem 3.8 is replaced by an integral. We consider only compact self-adjoint operators in this paper.

In the following key theorem, we provide a decomposition theorem for bivariate symmetric functions

**Theorem 3.8  (Product decomposition for symmetric bivariate functions)** *A symmetric function $f(\mathbf{x}, \mathbf{y}) \in L^2(\mathcal{X} \times \mathcal{X})$ corresponds to a compact self-adjoint operator, and can be decomposed as*

$$f(\mathbf{x}, \mathbf{y}) = \sum_{t=1}^{\infty} \lambda_t g_t(\mathbf{x}) g_t(\mathbf{y}),$$

*for some functions $g_t \in L^2(\mathcal{X})$, $\lambda_t \to 0$ as $t \to \infty$*

The above result gives a corresponding decomposition theorem (via Theorem C.1) for *partially symmetric functions in three variables*. Extending the result to beyond three variables would require extending this decomposition result for linear operators to the general multilinear operator case. Unfortunately, to the best of our knowledge, a decomposition theorem for multilinear operators is an open problem in the functional analysis literature. Indeed, even the corresponding discrete tensor case has only been studied recently. Instead, in the next section, we will use a result from *probability theory* instead, and obtain a proof for our decomposition theorem under additional conditions.

### 3.2.2 Argument via De Finetti's Theorem

In the previous section, we leveraged the interpretation of multivariate functions as multilinear operators. However, it is also possible to interpret multivariate functions as *measures on a product space*. Under appropriate assumptions, we will show that a De Finetti-like theorem gives us the required decomposition theorem for symmetric measures.

We first review De Finetti's theorem and related terms.

**Definition 3.9 (Infinite Exchangeability)** *An infinite sequence $X_1, X_2, \ldots$ of random variables is said to be exchangeable if for any $n \in \mathbb{N}$ and any permutation $\pi \in \mathcal{S}_n$,*

$$p(X_1, X_2, \ldots, X_n) = p(X_{\pi(1)}, X_{\pi(2)}, \ldots, X_{\pi(n)}) \tag{11}$$

We note that exchangeability as defined in the probability theory literature refers to symmetricity of the kind above, and is a distinct if related notion compared to that used in the rest of this paper.

Then, we have a class of De-Finetti-like theorems:

**Theorem 3.10 (De Finetti-like theorems)** *A sequence of random variables $X_1, X_2, \ldots$ is infinitely exchangeable iff, for all $n$, there exists a probability distribution function $\mu$, such that ,*

$$p(X_1, \ldots, X_n) = \int \Pi_{i=1}^n p(X_i; \theta) \mu(\mathrm{d}\theta) \tag{12}$$

*where $p$ denotes the pdf of the corresponding distribution*

This decomposes the joint distribution over $n$ variables into an integral over product distributions. De Finetti originally proved this result for 0-1 random variables, in which case the $p(X_i; \theta)$ are Bernoulli with parameter $\theta$ a real-valued random variable, $\theta = \lim_{n \to \infty} \sum_i X_i/n$. For accessible proofs of this result and a similar one for the case when $X_i$ are instead discrete, we refer the reader to [15, 2]. This result was later extended to the case where the variables $X_i$ take values in a compact set $\mathcal{X}$ by Hewitt and Savage [16]. (The proof in [16] first shows that the set of symmetric measures is a convex set whose set of extreme points is precisely the set of all product measures, i.e. independent distributions. Then, it establishes a Choquet representation i.e. an integral representation of this convex set as a convex combination of its extreme points, giving us a De Finetti-like theorem as above.) In this general case, the parameter $\theta$ can be interpreted as being *distribution*-valued – as opposed to real valued in the binary case described above. Our description of this result is terse for lack of space, see [2, pp.188] for details. Thus, we derive the following theorem:

**Theorem 3.11 (Product decomposition for Symmetric functions)** *Given an infinite sequence of documents with features $\mathbf{x}_i$ from a compact set $\mathcal{X}$, if a function $f : \mathcal{X}^m \to \mathbb{R}^+$ is symmetric in every leading subset of $n$ documents, and $\int f = M < \infty$, then $f/M$ corresponds to a probability measure and $f$ can be decomposed as*

$$f(\mathbf{x}) = \int \Pi_j g(\mathbf{x}_j; \theta) \mu(\mathrm{d}\theta) \tag{13}$$

*for some set of functions $\{g(\cdot; \theta)\}, g : \mathcal{X} \to \mathbb{R}$*

This theorem can also be applied to discrete valued features $X_i$, and we would obtain a representation similar to that obtained through tensor analysis in Section 3.1. Applied to features $X_i$

belonging to a compact set, we obtain the required representation theorem similar to the functional analytic theory of Section 3.2.1. However, note that De Finetti's theorem integrates over products of *probabilities*, so that each term is non-negative, a restriction not present in the functional analytic case. Moreover, we have an integral in the De Finetti decomposition, while via tensor analysis in the discrete case, we have a finite sum whose size is given by the rank of the tensor, and in the functional analytic analysis, the spectrum for bounded operators is discrete. De Finetti's theorem also requires the existence of infinitely many objects for which every leading finite subsequence is exchangeable. The similarities and differences between the functional analytic viewpoint and De Finetti's theorem have been previously noted in the literature, for instance in Kingman's 1977 Wald Lecture [19] and we discuss them further in Appendix E.

## 4  Experiments

For our experiments, we consider the information retrieval learning to rank task, where we are given a training set consisting of $n$ queries. Each query $q^{(i)}$ is associated with $m$ documents, represented via feature vectors $\mathbf{x}^{(i)} = (\mathbf{x}_1^{(i)}, \mathbf{x}_2^{(i)}, \ldots, \mathbf{x}_m^{(i)}) \in \mathcal{X}^m$. The documents for $q^{(i)}$ have relevance levels $\mathbf{r}^{(i)} = (r_1^{(i)}, r_2^{(i)}, \ldots, r_m^{(i)}) \in \mathcal{R}^m$. Typically, $\mathcal{R} = \{0, 1, \ldots, l-1\}$. The training set thus consists of the tuples $T = \{\mathbf{x}^{(i)}, \mathbf{r}^{(i)}\}_{i=1}^n$. $T$ is assumed sampled i.i.d. from a distribution $\mathcal{D}$ over $\mathcal{X}^m \times \mathcal{R}^m$.

**Ranking Loss Functionals**   We are interested in the NDCG ranking evaluation metric, and hence for the ranking loss functional, we focus on optimization-amenable listwise surrogates for NDCG; specifically, a convex class of strongly NDCG-consistent loss functions introduced in [24] and nonconvex listwise loss functions, ListNet [4] and the Cosine Loss. In addition, we impose an $\ell_2$ regularization penalty on $\|w\|$.

[24] *exhaustively* characterized the set of strongly NDCG consistent surrogates as Bregman divergences $D_\psi$ corresponding to strictly convex $\psi$ (see Appendix F). We choose the following instances of $\psi$: the Cross Entropy loss with $\psi(x) = 0.01(\sum_i x_i \log x_i - x_i)$, the square loss with $\psi(x) = \|x\|^2$ and the q-norm loss with $\psi(x) = \|x\|_q^2$, $q = \log(m) + 2$ (where $m$ is the number of documents). Note that the multiplicative factor in $\psi$ is significant as it does affect $\phi$.

**Ranking Functions**   The representation theory of the previous sections gives a functional form for listwise ranking functions. In this section, we pick a simple class of ranking functions inspired by this representation theory, and use it to *rerank* the scores output by various pointwise ranking functions. Consider the following class of exchangeable ranking functions $f(\mathbf{x})$ where the score for the $i$th document is given by:

$$f_i(\mathbf{x}) = b(\mathbf{x}_i)\Pi_{j \neq i} g(\mathbf{x}_i, \mathbf{x}_j; w) = b(\mathbf{x}_i)\Pi_{j \neq i} \exp\left(\sum_k w_k S_k(\mathbf{x}_i, \mathbf{x}_j)\right) \qquad (14)$$

where $b(\mathbf{x}_i)$ is the score provided by the base ranker for the $i$-th document, and $S_k$ are pairwise functions ("kernels") applied to $\mathbf{x}_i$ and $\mathbf{x}_j$. Note that $w = 0$ yields the base ranking functions. Our theory suggests that we can combine several such terms as $f_i(\mathbf{x}) = \sum_t b(\mathbf{x}_i; v_t)\Pi_{j \neq i} g(\mathbf{x}_i, \mathbf{x}_j; w_t)$. For our experiments, we only use one such term. A Gradient Boosting procedure can be used on top of our procedure to fit multiple terms for this series.

Our choice of $g$ is motivated by computational considerations: For general functions $g$, the computation of (14) would require $O(m)$ time per function evaluation, where $m$ is the number of documents. However, the specific functional form in (14) allows $O(1)$ time per function evaluation as $f_i(\mathbf{x}; w) = b(\mathbf{x}_i)\Pi_k(\exp(w_k \sum_{j \neq i} S_k(\mathbf{x}_i, \mathbf{x}_j)))$, where the inner term $\sum_{j \neq i} S_k(\mathbf{x}_i, \mathbf{x}_j)$ in the RHS does not depend on $w$ and can be precomputed. Thus after the precomputation step, each function evaluation is as efficient as that for a pointwise ranking function.

As the base pointwise rankers $b$, we use those provided by RankLib[1]: MART, RankNet, RankBoost, AdaRank, Coordinate Ascent (CA), LambdaMART, ListNet, Random Forests, Linear regression. We refer the reader to the RankLib website for details on these.

Table 1: Results for our reranking procedure across LETOR 3.0 datasets. For each dataset, the first column is the base ranker, second column is the loss function used for reranking.

| | OHSUMED | | TD2003 | | NP2003 | |
|---|---|---|---|---|---|---|
| | Base RankBoost | Reranked w/ Cross Ent | Base CA | Reranked w/ q-Norm | Base MART | Reranked w/ Square |
| ndcg@1 | 0.5104 | **0.5421** | **0.3500** | 0.3250 | 0.5467 | **0.5600** |
| ndcg@2 | 0.4798 | **0.4901** | 0.2875 | **0.3375** | 0.6500 | **0.6567** |
| ndcg@5 | 0.4547 | **0.4615** | 0.3228 | **0.3461** | 0.7112 | **0.7128** |
| ndcg@10 | 0.4356 | **0.4445** | 0.3210 | **0.3385** | 0.7326 | **0.7344** |

| | HP2003 | | HP2004 | | NP2004 | |
|---|---|---|---|---|---|---|
| | Base MART | Reranked w/ Cross Ent | Base RankBoost | Reranked w/ q-Norm | Base MART | Reranked w/ Square |
| ndcg@1 | 0.6667 | **0.7333** | 0.5200 | **0.5333** | 0.3600 | **0.3733** |
| ndcg@2 | 0.7667 | 0.7667 | 0.6067 | **0.6533** | 0.4733 | **0.4867** |
| ndcg@5 | 0.7546 | **0.7618** | 0.7034 | **0.7042** | 0.5603 | **0.5719** |
| ndcg@10 | 0.7740 | **0.7747** | 0.7387 | **0.7420** | 0.5951 | **0.6102** |

**Results**   We use the LETOR 3.0 collection [23], which contains the OHSUMED dataset and the Gov collection: HP2003/04, TD2003/04, NP2003/04, which respectively correspond to the listwise Homepage Finding, Topic Distillation and Named Page Finding tasks. We use NDCG as evaluation metric and show *gains* instead of losses, so larger values are better.

We use the following pairwise functions/kernels $\{S_k\}$: we *construct* a cosine similarity function for documents using the Query Normalized document features for each LETOR dataset. In addition, OHSUMED contains document similarity information for each query and the Gov datasets contain link information and a sitemap, i.e. a parent-child relation. We use these relations directly as the kernels $S_k$ in (14). Thus, we have two kernels for OHSUMED and three for the Gov datasets, and $w$ is 2- and 3-dimensional respectively. To obtain the scores $b$ for the baseline pointwise ranking function, we used Ranklib v2.1-patched with its default parameter values.

LETOR contains 5 predefined folds with training, validation and test sets. We use these directly and report averaged results on the test set. For the $\ell_2$ regularization parameter, we pick a $C$ from [0, 1e-5,1e-2, 1e-1, 1, 10, 1e2,1e3] tuning for maximum NDCG@10 on the validation set. We used gradient descent on $w$ to fit parameters. Though our objective is nonconvex, we found that random restarts did not affect the achieved minimum and used the initial value $w = 0$ for our experiments.

Since $w = 0$ corresponds to the base pointwise rankers, we expect the reranking method to perform as well as the base rankers in the worst case. Table 1 shows some results across LETOR datasets which show improvements over the base rankers. For each dataset, we compare the NDCG for the specified base rankers with the NDCG for our reranking method with that base ranker and the specified listwise loss. (Detailed results are presented in Appendix G). Gradient descent required on average only 17 iterations and 20 function evaluations, thus the principal computational cost of this method was the precomputation for eq. (14). The low computational cost and shown empirical results for the reranking method are promising and validate our theoretical investigation. We hope that this representation theory will enable the development of listwise ranking functions across diverse domains, especially those less studied than ranking in information retrieval.

## Acknowledgements

We acknowledge the support of ARO via W911NF-12-1-0390 and NSF via IIS-1149803, IIS-1320894, IIS-1447574, and DMS-1264033.

## Footnotes

[1]https://sourceforge.net/p/lemur/wiki/RankLib/

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
