[Supplementary Material]

## A  Prior Work on Listwise Ranking

The exchangeability assumption as defined in this paper on ranking functions seems intuitively natural, and indeed, specific ranking functions previously proposed in the literature are all exchangeable. While pointwise ranking functions are vacuously exchangeable, we now discuss two specifically listwise ranking functions previously proposed by [22] and [26] in light of our representation theory. These papers do not maintain a clear distinction between the ranking function and the loss, though their results do result in particular ranking functions.

Qin et al. [22] propose a Continuous Conditional Random Field model for "global ranking", which we call listwise ranking functions in this paper. They consider two ranking tasks: Pseudo Relevance Feedback and Topic Distillation. For Pseudo Relevance Feedback, inference on their continuous CRF boils down to the following linear-algebraic computation:

$$\hat{\mathbf{y}} = F(\mathbf{x}) = (\alpha^\intercal e I + \beta D - \beta S)^{-1} X \alpha \tag{15}$$

where $X$ represents the feature vectors for documents collated as a matrix, $S$ is the similarity matrix, $e$ is the all ones vector, $D = \sum_j S_{i,j}$, $\alpha$ and $\beta$ are learned parameters. $\beta = 0$ gives the linear ranking case. Then, for a permutation matrix $P$, the transformation $X \to PX$, $S \to PSP^{-1}$ implies that $F(P\mathbf{x}) = PF(\mathbf{x})$, i.e. this function is exchangeable. For Topic Distillation, inference on their continuous CRF corresponds to:

$$F(\mathbf{x}) = \frac{1}{\alpha^\intercal e}(2X\alpha + \beta(D_r - D_c)e) \tag{16}$$

where $X$ and $e$ are as above, $D_r$ and $D_c$ are derived from the link matrix indicating parent-child relationships between documents, $\alpha$, $\beta$ are learned parameters. Similar to the above, this can also be seen to be exchangeable.

Weston and Blitzer [26] propose Latent Structured Ranking for collaborative filtering, where they use document and query features explicitly and learn latent low rank representations for the query and document space. Letting $\mathbf{d}$ denote the set of all documents, $\{w_i\}$ the positional weights (to place more importance towards the top of the list), and $U, V, S$ being parameters to be learned, consider the scoring function:

$$f_{\text{lsr}}(q, \mathbf{d}) = \sum_{i=1}^{|\mathbf{d}|} w_i q^\intercal U^\intercal V \mathbf{d}_i + \sum_{i,j=1}^{|\mathbf{d}|} w_i w_j (\mathbf{d}_i^\intercal S^\intercal S \mathbf{d}_j) \tag{17}$$

The ranking function is defined as:

$$F_{lsr}(q) = \text{argmax}_{\mathbf{d}'} f_{lsr}(q, \mathbf{d}') \tag{18}$$

Since the ranking function is found by directly optimizing a listwise loss, it is listwise. While we don't have an explicit analytic form for this ranking function, the symmetry of (17) implies that it is exchangeable.

In general, any method which directly optimizes a listwise loss to perform inference on a new query, i.e. during the test phase, will result in a listwise ranking function. Both methods above fall under this regime. In some cases such as that of [22], this optimization problem results in a closed form expression. The key challenge these approaches face is that of performing fast inference, since an expensive operation such as solving an optimization problem or performing a matrix multiplication must be done for each test query. In our framework, test time inference is performed by evaluating the listwise ranking function. An important direction for future work is then to investigate classes of listwise ranking functions which can be efficiently evaluated. In this paper, we have proposed such a class in Equation 14.

## B  Proof of Theorem 3.7

In this section, we extend the symmetric tensor decomposition to the partially symmetric case. We first define the notion of a partially symmetric tensor:

**Definition B.1 (Partially symmetric tensor)** *An order-$k$ tensor $A$ is said to be partially symmetric w.r.t the first index iff for any $\sigma \in \mathcal{S}_{k-1}$*

$$A_{i_1 i_2 \ldots i_k} = A_{i_1 i_{(\sigma(1)+1)} \ldots i_{(\sigma(k-1)+1)}}, \tag{19}$$

*for any $(i_1, i_2, \ldots, i_k) \in \mathcal{X}^k$.*

We now extend Proposition 3.5 to provide a decomposition for *partially symmetric* tensors.

**Proposition B.2 (Pairwise decomposition for partially symmetric tensors)** *Any tensor $A$ partially symmetric in indices $i_2, i_3, \ldots, i_m$ can be decomposed in terms of order-2 tensors $V^j$ as*

$$A = \sum_j [V_{i_1, i_2}^j V_{i_1, i_3}^j, \ldots, V_{i_1, i_m}^j], \tag{20}$$

**Proof** For each fixed value of $i_1 = a \in \mathcal{X}$, Proposition 3.6 implies that there is an outer product decomposition, say $A_{a,\ldots} = \sum_j \otimes^k v_{a,j}$. Then, for each $j$, we can concatenate the $\{v_{a,j}\}_a$ into an order-2 tensor as $V_{i_1=a, i_2} = v_{a,i_2}$, and we will end up with a number of terms $\max_a rank(A_{a,\ldots})$. Since these ranks are all finite, the final decomposition is finite. ∎

Theorem 3.7 (decomposition for *partially symmetric* functions) then follows as the functional counterpart of this Proposition B.2 (decomposition for *partially symmetric* tensors); just as Proposition 3.6 (decomposition for *symmetric* functions) followed as the functional counterpart of Proposition 3.5 (decomposition for *symmetric* tensors).

## C  Extension to Partially Symmetric functions for compact $\mathcal{X}$

Similar to the discrete case, we can show that a decomposition theorem for multivariate symmetric functions leads to a corresponding pairwise partially symmetric decomposition theorem:

**Theorem C.1 (Product decomposition for partially symmetric functions)** *A partially symmetric function $h : \mathcal{X}^m \to \mathbb{R}$ symmetric in $\mathbf{x}_2, \ldots, \mathbf{x}_m$ on a bounded set $\mathcal{X}$ can be decomposed as*

$$h(\mathbf{x}_i, \{\mathbf{x}_{\setminus i}\}) = \sum_t \Pi_{j \neq i} g_t(\mathbf{x}_i, \mathbf{x}_j) \tag{21}$$

*for some set of functions $\{g_t\}_{t=1}^{\infty}, g_t : \mathcal{X} \times \mathcal{X} \to \mathbb{R}$*

**Proof** For a fixed $a$ and $\mathbf{x}_i = a$, we have from theorem 3.8, since $h(\mathbf{x}_i = a, \{\mathbf{x}_{\setminus i}\})$ is a symmetric function and $h(\mathbf{x}_i = a, \{\mathbf{x}_{\setminus i}\}) = \sum_t \Pi_{j \neq i} g_{t,a}(\mathbf{x}_j)$. We overload notation and define $g_t(\mathbf{x}_i = a, \mathbf{x}_j) = g_{t,a}(\mathbf{x}_j)$ and obtain a pairwise decomposition for $h$. ∎

## D  Proof of Theorem 3.8

**Theorem D.1 (Symmetric Product decomposition for symmetric pairwise functions)** *A continuous symmetric function $f(x, y) \in L^2(\mathcal{X} \times \mathcal{X})$ corresponds to a compact self-adjoint operator $A$, and can be decomposed as*

$$f(x, y) = \sum_t \lambda_t g_t(x) g_t(y),$$

*for some functions $g_t \in L^2(\mathcal{X})$, $\lambda_t \to 0$ as $t \to \infty$*

**Proof** A continuous symmetric function $f(x, y) \in L^2(\mathcal{X} \times \mathcal{X})$ corresponds to a Hilbert-Schmidt linear operator: $(Ag)(x) = \int f(x, y)g(y)d\mu(y)$. From Theorem VI.23 in [25] on Hilbert-Schmidt operators, $A$ is compact. It can be easily shown that $A$ is symmetric; since $f$ is $L^2$, $A$ is also bounded; so that it follows that $A$ is a bounded self-adjoint operator. Thus, the Hilbert Schmidt theorem applies, and we obtain a spectral decomposition for $A$. Noting that $f$ in turn can be expressed in terms of the operator as $f(x, y) = \langle x, A(y) \rangle = \langle A(x), y \rangle$, yields the corresponding decomposition for $f$ as

$$f(x, y) = \sum_t \lambda_t \phi_t(x) \phi_t(y),$$

where $\{\phi_n\}$ is a complete orthonormal basis for $L^2(\mathcal{X})$ and $\lambda_t \to 0$ as $t \to \infty$ ∎

# E  Discussion of the relation between the different analyses

Some remarks on the similarities and contrast between the tensor, functional analytic, and probabilistic viewpoints via De-finetti's theorem:

**I.** De Finetti's theorem requires the existence of infinitely many documents for which every leading subsequence is exchangeable and simple counterexamples can be given for finite exchangeable sequences[19]. Consequently, Diaconis and Freedman have considered Finite Exchangeable Sequences, and show that while no De Finetti-like theorem can hold for finite exchangeable sequences, the total variational distance between the De-Finetti representation and the true joint distribution is $O(\frac{1}{n})$[13, 12].

**II.** The RHS of the theorem contains an integral over products of probabilities and so, each term on the RHS is non-negative. Jaynes[18] points out that this non-negativity is the reason why finite versions of the theorem do not hold. Jaynes gives a proof for 4 variables showing that for finite exchangeable sequences, a decomposition always exists if negative terms are permitted. This proof is essentially a restricted version of that of section 3.1 using tensor decomposition. [2] Jaynes also points out a relationship between our problem and the N-representability problem which arises in physics, which we will investigate in future work.

**III.** Even in the discrete case, De Finetti's theorem leads to an integral, while the tensor analysis leads us to a finite sum whose size is given by the rank of the matrix. Similarly, in the functional analytic literature, the spectrum for bounded operators is discrete while that for unbounded operators has an integral form.

**IV.** The measure $\mu$ in this theorem is independent of the $n$ considered, i.e. unlike the previous theorems which would apply to particular choices of $n$, this version relates decompositions across *different* $n$'s.

# F  NDCG consistent loss functions

[24] *exhaustively* characterized the set of strongly NDCG consistent surrogates. First recall the definition of a Bregman divergence $D_\psi$ corresponding to a strictly convex $\psi$ as:

$$D_\psi(p, q) = \psi(p) - \psi(q) - \langle \nabla \psi(p), p - q \rangle. \tag{22}$$

A function $h : \mathbb{R}^m \to \mathbb{R}^m$ is called order preserving if $x_i \succeq x_j \Rightarrow h(x_i) \succeq h(x_j)$. [24] then showed that the set of strongly NDCG-consistent convex surrogates can be completely characterized as:

$$\phi(\mathbf{s}, \mathbf{r}) = D_\psi \left( \frac{G(\mathbf{r})}{\|G(\mathbf{r})\|_D}, (\nabla \psi)^{-1}(\mathbf{s}) \right), \tag{23}$$

for some strictly convex $\psi$ with $\nabla \psi$ order preserving, and where $G(t) = \log(1 + t)$ and $\|\mathbf{r}\|_D = \max_{\pi \in \mathcal{S}_m} \sum_{j=1}^m \frac{|r_i|}{F(\pi(k))}$. $\mathbf{r}, \mathbf{s}$ are the score vectors associated with the true and predicted rankings respectively. $\mathbf{r}$ is a vector of relevance labels for documents in a list, and $\mathbf{s}$ are the predicted scores from the ranking function, which can be sorted to yield the predicted ranking.

Given a set of ranking functions $\mathcal{F} = \{f : \mathcal{X}^m \to \mathbb{R}^m\}$, this surrogate loss can be used to find the optimal ranking function over $\mathcal{D}$ as:

$$f^* = \arg\min_{f \in \mathcal{F}} \mathbb{E}_\mathcal{D}[\phi(\mathbf{s}, f(\mathbf{x}))] \tag{24}$$

# G  Experimental Details

Tables 2 through 8 tabulate our results on various datasets. We found that the choice of loss function does not change the results significantly, so results for each base ranker with only one loss function are shown. With the default parameter settings for RankLib, some methods such as LambdaMART and the ListNet base ranker overfit heavily and are excluded from these results.

### Table 2: Results on OHSUMED

| | Base RankBoost | Reranked w/ Cross Ent | Base MART | Reranked w/ Cross Ent | Base ListNet | Reranked w/ Cosine Loss |
|---|---|---|---|---|---|---|
| ndcg@1 | 0.5104 | **0.5421** | 0.4760 | 0.4760 | **0.4434** | 0.4339 |
| ndcg@2 | 0.4798 | **0.4901** | 0.4065 | **0.4176** | 0.4641 | **0.4729** |
| ndcg@5 | 0.4547 | **0.4615** | 0.3842 | **0.3939** | 0.4327 | **0.4331** |
| ndcg@10 | 0.4356 | **0.4445** | **0.3677** | 0.3671 | 0.4223 | **0.4237** |

### Table 3: Results on HP2003

| | Base Ranker MART | Reranked with NDCG Cross Ent |
|---|---|---|
| ndcg@1 | 0.6667 | **0.7333** |
| ndcg@2 | 0.7667 | 0.7667 |
| ndcg@5 | 0.7546 | **0.7618** |
| ndcg@10 | 0.7740 | **0.7747** |

### Table 4: Results on HP2004

| | Base Ranker RankBoost | Reranked with NDCG q-Norm | Base Ranker Random Forests | Reranked with NDCG q-Norm |
|---|---|---|---|---|
| ndcg@1 | 0.5200 | **0.5333** | 0.5467 | 0.5467 |
| ndcg@2 | 0.6067 | **0.6533** | 0.6400 | **0.6533** |
| ndcg@5 | 0.7034 | **0.7042** | 0.6795 | **0.6938** |
| ndcg@10 | 0.7387 | **0.7420** | **0.7157** | 0.7136 |

### Table 5: Results on TD2003

| | Base Ranker Coordinate Ascent | Reranked with NDCG q-Norm | Base Ranker Linear regression | Reranked with ListNet Loss |
|---|---|---|---|---|
| ndcg@1 | **0.3500** | 0.3250 | 0.3200 | **0.3600** |
| ndcg@2 | 0.2875 | **0.3375** | 0.3000 | **0.3100** |
| ndcg@5 | 0.3228 | **0.3461** | 0.2916 | **0.2957** |
| ndcg@10 | 0.3210 | **0.3385** | **0.3193** | 0.3141 |

### Table 6: Results on TD2004

| | Base Ranker Linear regression | Reranked with ListNet Loss | Base Ranker Random Forests | Reranked with NDCG Square |
|---|---|---|---|---|
| ndcg@1 | 0.2000 | 0.2000 | 0.5600 | 0.5600 |
| ndcg@2 | 0.2667 | **0.3000** | 0.4667 | **0.4867** |
| ndcg@5 | 0.2736 | **0.2979** | 0.3903 | **0.3939** |
| ndcg@10 | 0.2545 | **0.2616** | 0.3531 | **0.3546** |

### Table 7: Results on NP2003

| | Base Ranker MART | Reranked with NDCG Square | Base Ranker Random Forests | Reranked with ListNet Loss |
|---|---|---|---|---|
| ndcg@1 | 0.5467 | **0.5600** | 0.6000 | 0.6000 |
| ndcg@2 | 0.6500 | **0.6567** | 0.7167 | **0.7233** |
| ndcg@5 | 0.7112 | **0.7128** | 0.7673 | **0.7697** |
| ndcg@10 | 0.7326 | **0.7344** | **0.8017** | 0.8016 |

### Table 8: Results on NP2004

| | Base Ranker MART | Reranked with NDCG Square | Base Ranker Linear regression | Reranked with ListNet Loss |
|---|---|---|---|---|
| ndcg@1 | 0.3600 | **0.3733** | 0.3000 | **0.3167** |
| ndcg@2 | 0.4733 | **0.4867** | 0.4583 | 0.4583 |
| ndcg@5 | 0.5603 | **0.5719** | 0.5857 | **0.5878** |
| ndcg@10 | 0.5951 | **0.6102** | 0.6390 | **0.6468** |

# H Directions for Future Work

We hope that this representation theory will enable the development of listwise ranking functions across diverse domains, especially those less studied than ranking in information retrieval.

The analysis via De Finetti's theorem provides a Bayesian perspective to this problem, where a ranking function specifies a distribution over the objects. In future work, these probabilistic connections could be used to devise novel exchangeable listwise ranking functions. We would also like to develop a representation theory for permutation-valued functions which take several ranked lists as input, corresponding to the rank-aggregation problem. This more accurately models the learning to rank setting in information retrieval where document features in typical datasets are themselves the outputs of ranking functions.

While our assumption of exchangeability for ranking functions was very natural, we note that there might be also cases of interest when ranking functions might not be exchangeable. For instance, when the objects have a particular sequential (e.g. temporal) or spatial organization (e.g. via a graph). In such cases, we might need to first embed the set of objects into a vector space before considering the assumption of exchangeability; we plan to investigate these and other principled approaches to such problems in future work.

---

of the author. We hope that, with its publication, the useful results of this representation will become more readily obtainable". To the best of our knowledge, this work was not published.

## Footnotes

[2]Jaynes[18] also discusses an extension to the compact $\mathcal{X}$ case, "A more powerful and abstract approach, which does not require us to go into all that detail, was discovered by Dr. Eric Mjolsness while he was a student