[Reviews · NeurIPS 2014]

Submitted by Assigned_Reviewer_17

The authors propose to formalize "the notion that the ranking function depends only on the object features, and not the order in which the documents are presented." This is a good idea, but the proposed notion of exchangeability is too strict in my opinion: we can capture the intended notion without the strict equality in eqn 1 and 2. We just want the order of the scores to be preserved, not their exact values. In terms of clarity, there are sections that are quite unclear, as pointed out below.

You should define symmetric function in the proof of Thm 3.2.

You should define p in Def 3.9. You probably don't mean to use p on both sides of the equality.

In Thm 3.11, it's not clear what values \theta takes.

In Thm 3.10, it's not clear what is meant by the integral over \theta, when later it says that \theta is a random variable.
Summary: The idea is good, but this paper is not clear enough in its main result (Thm 3.11).

After author feedback: I realize that 3.10 is not the main result: I meant 3.11. The same criticism remains unaddressed in your rebuttal: what do you mean by the integral over \theta, when you say elsewhere that \theta is a real-valued random variable.

Submitted by Assigned_Reviewer_30

The paper considers settings where the goal is to learn a permutation-valued function, such as in subset ranking/information retrieval applications. Specifically, it focuses on settings where one learns a vector-valued function that assigns a real-valued score to each document and then sorts the objects according to these scores. E.g. if there are m objects to rank, represented as m feature vectors x_1, …, x_m, then one frequently learns a weight vector w and sorts the objects according to scores w.x_1, …, w.x_m. The paper refers to this as a "pointwise" approach, since the score assigned to x_i does not depend on the feature vectors for the remaining objects x_j, and advocates learning directly a function that collectively maps m feature vectors x_1,…,x_m to m scores s_1,…,s_m.

The primary contribution is in developing a mathematical characterization of classes of such "collective scoring" functions that satisfy a natural symmetry/exchangeability property, which simply says that if we exchange two feature vectors x_i and x_j, then the scores s_i and s_j assigned by the function to the corresponding objects should also be exchanged accordingly. The characterization involves tools from tensor analysis when there is a finite set of possible feature vectors and De Finetti-like theorems in the general case.

The approach is interesting overall, but my main concern is that there is no clear evidence in the paper for why it is useful. In particular, the experiments could have compared methods that learn standard linear "pointwise" functions with methods that learn functions from exchangeable function classes as proposed (keeping other parameters, such as loss function minimized, constant); this would have shown clearly what sorts of benefits might be offered by learning functions in the proposed form of function classes. Instead, the paper contains experiments which start with a baseline (linear?) function learned by some standard method, incorporate this baseline function in the "exchangeable" function class, and then re-learn a scoring function from this class. The experiments are "brute-force" in style (lots of data sets, baseline linear learning methods/loss functions), without much clear insight into what is being tested or what are the precise benefits of the proposed approach.

Small comment: The title is too broad for this work (both "representation theory" and "ranking functions" have other meanings in different mathematical contexts); a better title would be something like "Symmetric (or exchangeable) permutation-valued functions with applications in …"
Summary: The paper considers learning from more general classes of permutation-valued functions than what are generally used in current subset ranking/IR applications. Interesting approach, but needs further development and validation; in particular experiments need to be more clearly thought through.

Submitted by Assigned_Reviewer_42

This paper considers the problem of learning rankings from features. They develop listwise ranking functions and show that under an assumptions of exchangability a nice formulation of the loss functions can be given. The authors then develop some representation theory for rank functions followed by examples of rank functions that satisfy the theoretical requirements. The authors close with some empirical results.

I thought this paper was a very clever use of DeFennitti's theorem. I really liked Theorem 3.2. The empirical results are reasonable.
The basic idea in this paper is very good, and one complaint could be that once you realize the idea the rest is obvious but since no one had before this makes this paper very good.
Summary: This paper considers the problem of learning rankings from features. They develop listwise ranking functions and show that under an assumptions of exchangability a nice formulation of the loss functions can be given.

Submitted by Assigned_Reviewer_43

SUMMARY

The paper derives a representation theorem for the class of "exchangeable" listwise scoring functions employed in learning to rank problems. The results show that for finite instance spaces, such functions are expressible as a particular combination of similarity functions over pairs of instances. For compact spaces, a similar result holds for the case of lists of size two; for general lists, the result holds under additional assumptions by appeal to deFinetti's representation theorem.

SIGNIFICANCE

The issue of designing listwise loss functions has been well studied, but the design of listwise scoring functions less so. As noted in the paper, some works focus (e.g. [4]) involve scoring each instance within the list with a single function, albeit one that is optimised in a listwise sense. The results in the paper are interesting, and while largely intuitive not a-priori obvious. The stated goal is to give a recipe with which to guide the design of listwise ranking functions. I think the observation that this can be done by designing similarity functions on pairs of instances, and suitably combine them, is a good first step towards this goal.

All results rely on a symmetry assumption on the listwise ranking function. This seems reasonable, and minimally holds e.g. for the pointwise case. The results for the compact case in 3.2.2 additionally rely on the scoring function being an unnormalised probability density over the instances. It is less clear to me how reasonable this is. The requirement of nonnegativity of the scoring function could be handled by an appropriate link function (e.g. exp(.) as done in the experiments). But this would require suitable interplay with the surrogate loss being optimised to ensure convexity, something that may deserve some comment.

The experiments show that it is possible to post-process the outputs of a pointwise ranker (even one trained to optimise a listwise loss) using the representation theorem. The experiments are the weakest part of the paper, and likely more could be done to make a convincing case for the value of the recipe (e.g. try a wider class of pairwise functions, have a comparison of pairwise "as-is" versus pointwise to check the perils of overfitting, et cetera). But good performance in this regime is still interesting, as (to my knowledge) improving upon methods trained to optimise listwise losses is non-trivial. So I think they are indicative of the promise of the representation theorem in guiding the design of scoring functions.

QUESTIONS/COMMENTS

- The statement of Theorem C.1 apparently holds for m > 2 instances in the list. This seems to not agree with the text; perhaps the point is that existence of a multilinear symmetric decomposition would imply the result? If so the appeal to Theorem 3.8 in the proof seems misleading. It does seem like you could have such a corollary to 3.11, so perhaps that should be stated.

- Even if a simple consequence of 3.10, I think a proof of Theorem 3.11 should be included in the appendix. The statement of the theorem alone leaves some questions:
* There is an implicit "for every m" needed? In which case, f should actually be defined for every possible m in X^m?
* In 3.10 the "g" function is (I presume) the marginal, derived appropriately from the joint p. Does a similar restriction exist here in terms of the function g?
* The requirement of an infinite sequence in deFinetti must of course carry over to the theorem, but it is a little hard to understand what exactly it is requiring of the ranking function and/or the input space. The comment about the result requiring "infinitely many objects for which every leading finite subsequence is exchangeable" could be expanded. Isn't this implicit when one assumes that f is symmetric in its arguments, and hence exchangeable regardless of the provided inputs (which could be one of infinitely many possible values, since X is infinite)?
* What does it mean for theta to be a "random distribution" in the prelude to the theorem?
- The extension of 3.11 to the partially symmetric case seems non-obvious, and minimally requires a careful statement of the infinite sequence assumption, and the appropriate boundedness of the integral of f. I would recommend making this explicit.

OTHER COMMENTS

- I like that the paper builds up from the simple case of finite instance spaces to a more general theory.

- Line 211, for three documents f_i : X^3 \to R, so f_i is a tensor? Or do you group the last two elements into one?

- Remark II pg 5, tensor completion seems reasonable for finite spaces and suitably large training sets, but in practice it seems likely that one would face cold-start problems (no observations for a particular "row") due to limited coverage of the instance space.

- Line 268, is the corresponding decomposition theorem referred to Theorem C.1?

- It might be better to include Theorem C.1 in the body and defer 3.8 to the appendix if space is an issue.

- Appendix F is unclear. What are r, s? How exactly is the surrogate loss used for optimisation?

- The paper is generally well written. There were some small typos worth addressing:
* Def 2.1, \to instead of \mapsto in definition of f_j.
* Missing period line 169.
* Period instead of comma line 186.
* Theorem 3.8 proof, missing boldface in x, y.
* Theorem 3.11, we need f : X^m \to R_+?
* Fix capitalisation in ref [3].

- Other comments:
* Why the use of braces in { x_{\i} }?
* Include summation indices in Prop 3.5, 3.6, Thm 3.8?
* Def 3.4, use \pi rather than \sigma for consistency?
* Theorem C.1 proof should come after 3.8.
* Notation clash in (14), S for training set and S_k for similarity function?
Summary: The paper derives a representation theorem for the class of "exchangeable" listwise scoring functions employed in learning to rank problems. The theoretical results are interesting, and while largely intuitive not a-priori obvious. The experimental results, while far from conclusive, are indicative of the promise of the approach.
Author Feedback
Author rebuttal: The reviewers have widely varying opinions on our paper. While AR42 has given it a very positive review, AR17 and AR30 have given it significantly worse reviews. We believe this is because AR17 has a misunderstanding regarding which result is the principal novelty of the paper. AR30 raises a point regarding our experiments, but as we explain below, our experiments encapsulate the condition (s)he proposes. We thank the reviewers for their valuable suggestions, and believe that they recognize the novelty of our results modulo the above misunderstandings; we are thus very hopeful that they would increase their scores in light of our response below.

AR17:

AR17's main objection is that "the paper is not clear enough in its main result (Thm 3.10)".

We would like to clarify that Theorem 3.10 is not our main result, but is De Finetti's theorem, a well-known result from the literature.

AR17 also points out that details such as what values \theta takes in Theorem 3.10 are described only briefly. Since it was a description of the known result of De Finetti's theorem, it was necessarily terse due to space constraints, where we referred the reader to [2] for additional details.

We thank the reviewer for their general suggestion to make the paper clearer, but feel their judgement that the paper be rejected on these grounds is a bit too harsh. Please also see the other two reviews for a summary of our contributions, which agree that the paper presents substantially novel and interesting theoretical developments.

We are hopeful that the reviewer will modify their review in light of the above explanation.

AR30:

AR30 believes our experiments are weak. We would like to clarify our insight into what is being tested:

* The goal of our experiments is to measure the improvement that can be obtained by reranking the output of *state-of-the-art* pointwise ranking procedures using our listwise ranking functions. Our baseline functions are not just linear, but are *state-of-the-art* pointwise functions, such as MART, RankBoost etc (described in lines 365-367).

* The reviewer proposes that we compare linear pointwise functions with a method that directly learns exchangeable function classes. Our reranking experiments are very much in the spirit of what the reviewer proposes, except that our exchangable ranking function is learnt via a two stage procedure, first setting its "pointwise component" b() equal to the learnt pointwise ranking function (that is being re-ranked), and then fitting the pairwise terms.

What the reviewer is proposing is to also learn the pointwise component from scratch. Incidentally, we do have experiments with this comparison, and will be glad to add these to the appendix. Here, our exchangeable ranking function again handily beats the pointwise ranking function. But statistically this is not that interesting a comparison: note that our function class is strictly more general than pointwise ranking functions (it includes the latter, which can be seen by setting the pairwise term coefficients to zero); accordingly, statistically, it is not at all surprising for our approach to beat the linear pointwise ranking functions. Incidentally, an earlier submission of an earlier version of this paper in another conference, had precisely the experiments suggested by the reviewer: and the reviews there were unanimous that this was an extremely theoretically interesting paper, except that the experiments were not statistically surprising, and that it could have been accepted if it had had a re-ranking experiment (as in our current submission) instead! (We note that for space constraints, we cannot include both these kinds of experiments!)

* Our experiments certainly do involve many datasets and base pointwise ranking functions, but this was with the aim of being exhaustive in showing that across such settings, our proposed reranking method shows improvements over the base pointwise functions.

In light of the above explanation, and the reviewer's own assessment that the paper presents interesting and important theoretical developments, we are sincerely hopeful they would modify their score.

AR42:

We thank the reviewer for their kind comments.